

# Sporting tournaments and changed birth rates 9 months later: a systematic review

Gwinyai Masukume[1], Victor Grech[2] and Margaret Ryan[3]

[1] School of Public Health, Physiotherapy and Sports Science, University College Dublin, Dublin, Ireland
[2] Academic Department of Paediatrics, Medical School, Mater Dei Hospital, Msida, Malta
[3] Trinity College Dublin, Dublin, Ireland

## ABSTRACT

**Introduction:** Major sporting tournaments may be associated with increased birth rates 9 months afterwards, possibly due to celebratory sex. The influence of major sporting tournaments on birth patterns remains to be fully explored.

**Methods:** Studies that examined the relationship between such events and altered birth metrics (number of births and/or birth sex ratio (male/total live births)) $9(\pm1)$ months later were sought in PubMed and Scopus and reported *via* standard guidelines. Database searches were conducted up to 7 November 2022.

**Results:** Five events led to increased birth metrics $9(\pm1)$ months later and these included the Super Bowl, the 2009 UEFA Champions League, the 2010 FIFA World Cup, the 2016 UEFA Euros and the 2019 Rugby World Cup. Several *la Liga* soccer matches also had effects. With a few exceptions, major American football, Association football (soccer) and Rugby apex tournaments in Africa, North America, Asia and Europe were associated with increases in the number of babies born and/or in the birth sex ratio $9(\pm1)$ months following notable team wins and/or hosting the tournament. Furthermore, unexpected losses by teams from a premier soccer league were associated with a decline in births 9 months on.

**Conclusions:** This systematic review establishes that major sporting tournaments have a notable impact on birth patterns, influencing both birth rates and sex ratios. Emotional intensification during these events likely triggers hormonal shifts, driving changes in sexual activity and subsequently shaping birth rates, often positively, about 9 months later. The context is crucial, especially when a region/country hosts a major single-sport tournament or participates for the first time, as population excitement is likely to be at its peak. These findings hold significance for healthcare planning and highlight the role of societal events in shaping demographic trends.

**PROSPERO registration:** CRD42022382971.

## INTRODUCTION

A rise in the number of births nine months later has been linked, in some settings, to a variety of events, including carnivals, religious and secular holidays such as Christmas and New Year's Day (January 1) (*Kadhel et al., 2017*; *Macfarlane et al., 2019*; *Régnier-Loilier & Rohrbasser, 2011*). This increase has been attributed to the holiday celebrations

Corresponding author
Gwinyai Masukume,
parturitions@gmail.com

encouraging more conception, nine months earlier (*Jukic et al., 2013*), through increased sexual activity among the populace.

Sports are a dominant part of culture around the world, with major sporting tournaments attracting millions of attendees and even billions of viewers (*Maennig & Zimbalist, 2012*). A major sporting tournament, led by specialized authorities, brings notable impacts and media coverage, influencing economic, tourism, and infrastructure aspects for host communities. Such events often feature accompanying festivities and cultural programs. Examples include the Olympic Games, *Fédération Internationale de Football Association* (FIFA) World Cup and the Super Bowl (*Byers, Slack & Parent, 2012*). It is important to consider how the outcomes of these tournaments affect population-level demographics because fans often have a deep investment in the triumph of their beloved teams, which may influence fertility (*Gorelik & Bjorklund, 2015*). Emerging evidence suggests that major sporting tournaments are also associated with increased birth rates nine months afterwards (*Grech & Zammit, 2019*; *Masukume & Grech, 2015*; *Masukume, Grech & Scherb, 2016*; *McKenna, Znaczko & Morrison, 2019*; *Montesinos et al., 2013*). For instance, in Northern Ireland (NI), nine months after the 2016 Union of European Football Associations (UEFA) Euros Championship, in which NI was competing for the first time, a 2% rise in births was observed in March 2017 (*McKenna, Znaczko & Morrison, 2019*). Nine months following Futbol Club Barcelona's victory in the 2009 UEFA Champion's league, there was a considerable 16% increase in births in Catalonia (*Montesinos et al., 2013*). On the other hand, it has been reported that nine months following the United States (US) National Football League's (NFL) Super Bowl, the number of births in winning cities did not rise (*Hayward & Rybińska, 2017*). However, the US sex ratio at birth (SRB), defined as male divided by total live births, increased nine months after several Super Bowls (*Grech & Zammit, 2019*). Nine months after the 2010 FIFA World Cup, there were over 1,000 more live births, and the SRB was higher (*Masukume & Grech, 2015*; *Masukume, Grech & Scherb, 2016*). In addition, there have been several anecdotal media stories about baby booms nine months after sporting tournaments, for instance a boom in Iceland after the 2016 UEFA Euros Championship, but subsequent statistical analyses of birth data has not supported these claims (*Grech & Masukume, 2017*). In contrast to the aforementioned, but consistent with the paradigm, unexpected losses of Spanish local soccer teams have been reported to lead to a 0.8% decrease in the local number of births nine months later (*Bernardi & Cozzani, 2021*).

In light of the apparent scarcity of recent studies initially identified, our research aims to explore and fill a gap in the literature. Given the sometimes conflicting and inconsistent reports, our objective was to conduct a systematic review of the literature evaluating the relationship between sporting events and live birth rate/proportion changes 9(±1) months later. If sporting events alter birth patterns significantly, this could have an impact on the need for midwifery, medical and other healthcare personnel as well as resources.

## METHODS

The protocol for this systematic review adheres to the Preferred Reporting Items for Systematic Reviews and Meta-Analyses (PRISMA) standards (*Page et al., 2021*). This

review was registered prospectively on the International prospective register of systematic reviews site PROSPERO, CRD42022382971. This step was taken subsequent to conducting initial searches and piloting the study selection process. The formal screening of search results against eligibility criteria followed suit, aligning with PROSPERO registration guidelines due to the substantial progress in our review's advancement. Portions of this text were previously published as part of a preprint (*Masukume, Grech & Ryan, 2022*).

### Search strategy

A search strategy and search terms were developed for this study. Without regard to language, a systematic literature search was conducted. Using the search strategy (Table 1), potentially relevant studies were looked up from database inception through to 7 November 2022 in the electronic databases PubMed and Scopus. The first author carried out all searches. The following search term syntax was used:

PubMed: *"Sports"[Mesh] AND ("Birth Rate"[Mesh] OR "Sex ratio"[MESH] OR "Fertility"[MESH])*

Scopus: *KEY ("sports") AND KEY ("birth") OR KEY ("birth rate") OR KEY ("fertility")*

Using Covidence systematic review software (www.covidence.org), we completed the title/abstract and full-text screening for this study. This was performed independently by two authors (GM and MR). The reference lists of the studies that were included in the search were manually searched as an add-on. The 'cited by' function of Google Scholar was also used to find articles citing the included studies.

### Eligibility

Table 2 lists the inclusion and exclusion criteria that were employed in this study.

### Data extraction and analysis

Data were extracted from the included studies by two reviewers (GM and MR). Author name and year, country or region of the tournament, sport and change in birth pattern (total births/SRB) were extracted. Extraction differences were discussed by these two authors and were resolved. Due to the variety of sports, eras and geographic locations, a narrative synthesis was done in accordance with the Synthesis Without Meta-analysis (SWiM) guidelines (*Campbell et al., 2020*).

### Quality assessment

As far as we are aware, there is no instrument that has been validated for use in evaluating the quality of ecological studies (*Betran et al., 2015*). However, the group exposure (sporting tournament) was assessed according to how the population that experienced the tournament defined it; in other words, there was common understanding that the sporting tournament was significant, and the studies characterized it as such. There was minimal concern about outcome bias because the human live birth statistics of the entire population (and not just a sample) was available and the danger of misclassifying sex at birth is low (*Davis, Gottlieb & Stampnitzky, 1998*).

**Table 1 Search terms used to identify relevant studies.**

| Exposure | Outcome |
|---|---|
| **PubMed** | |
| MeSH term "Sports" | MeSH term "Birth rate" |
| Athletic performance | Birth rates |
| Baseball | Rate, Birth |
| Basketball | Natality |
| Bicycling | Natalities |
| Boxing | Fertility rate |
| Cricket sport | Fertility rates |
| Football | Rate, fertility |
| Golf | Age-specific birth rate |
| Gymnastics | Age specific birth rate |
| Hockey | Age-specific birth rates |
| Martial arts | Birth rate, age-specific |
| Mountaineering | Age-specific fertility rate |
| Racquet sports | Age specific fertility rate |
| Return to sport | Age-specific fertility rates |
| Rugby | Fertility rate, age-specific |
| Running | Rate, age-specific fertility |
| Skating | MeSH term "Sex Ratio" |
| Snow sports | Ratio, sex |
| Soccer | Ratios, sex |
| Sports for persons with disabilities | Sex ratios |
| Team sports | MeSH term "Fertility" |
| Track and field | Fecundability |
| Volleyball | Fecundity |
| Walking | Differential fertility |
| Water sports | Fertility, differential |
| Weight lifting | Fertility determinants |
| Wrestling | Determinant, fertility |
| Youth sports | Determinants, fertility |
| | Fertility determinant |
| | Subfecundity |
| | Fertility preferences |
| | Fertility preference |
| | Preference, fertility |
| | Preferences, fertility |
| | Fertility, below replacement |
| | Below replacement fertility |
| | Marital fertility |
| | Fertility, marital |
| | Natural fertility |
| | Fertility, natural |

| Exposure | Outcome |
|---|---|
| **PubMed** | |
| | World fertility survey |
| | Fertility survey, world |
| | Fertility surveys, world |
| | Survey, world fertility |
| | Surveys, world fertility |
| | World fertility surveys |
| | Fertility incentives |
| | Fertility incentive |
| **Scopus** | |
| KEY field "Sports" | KEY field "births" |
| | KEY field "birth rate" |
| | KEY field "fertility" |

Note:
MeSH, Medical Subject Headings. The Scopus field code KEY was used in an advanced search. The KEY field code is a combined code which searches AUTHKEY, INDEXTERM, TRADENAME and CHEMNAME fields. AUTHKEY (key words assigned to the document by the author), INDEXTERM (controlled vocabulary terms assigned to the document), TRADENAME (a name used to identify a commercial product or service) and CHEMNAME (chemical name).

**Table 2 Inclusion and exclusion criteria.**

| | Inclusion | Exclusion |
|---|---|---|
| Participants | Population exposed to sporting tournament | No sporting tournament |
| Exposure | Sporting tournament* | No sporting tournament |
| Comparison | Control population from immediate prior or future years (same time of the year) in same geographic region not exposed to sporting tournament | Study without this outcome |
| Outcome | Change in total births and/or sex ratio 9(±1) months later | Study without this outcome |
| Study design | Ecological | Letters, editorials, reviews, *etc.* |
| Language | All languages | Not applicable |
| Date | Inception of database | 7 November 2022 |

Note:
* Direct observation of the tournament or indirectly *via* the media.

## RESULTS

The search strategy was used to find 111 and 332 articles from PubMed and Scopus respectively. After uploading the search results into Covidence and removing 12 duplicates the authors (GM and MR) evaluated each of the remaining 431 articles' titles and abstracts individually (Fig. 1). After examining all titles and abstracts independently, these authors convened to discuss and resolve any differences. All differences were resolved. Six articles remained (Fig. 1). An additional four articles were found by citation searching. The final data extraction included all 10 of the articles that made it to the full-text screening stage. The characteristics of sporting events with altered birth rates or SRB 9(±1) months later are listed in Table 3. Major American football, Association football (soccer) and rugby championships were linked to increases in the number of births and/or birth sex ratio
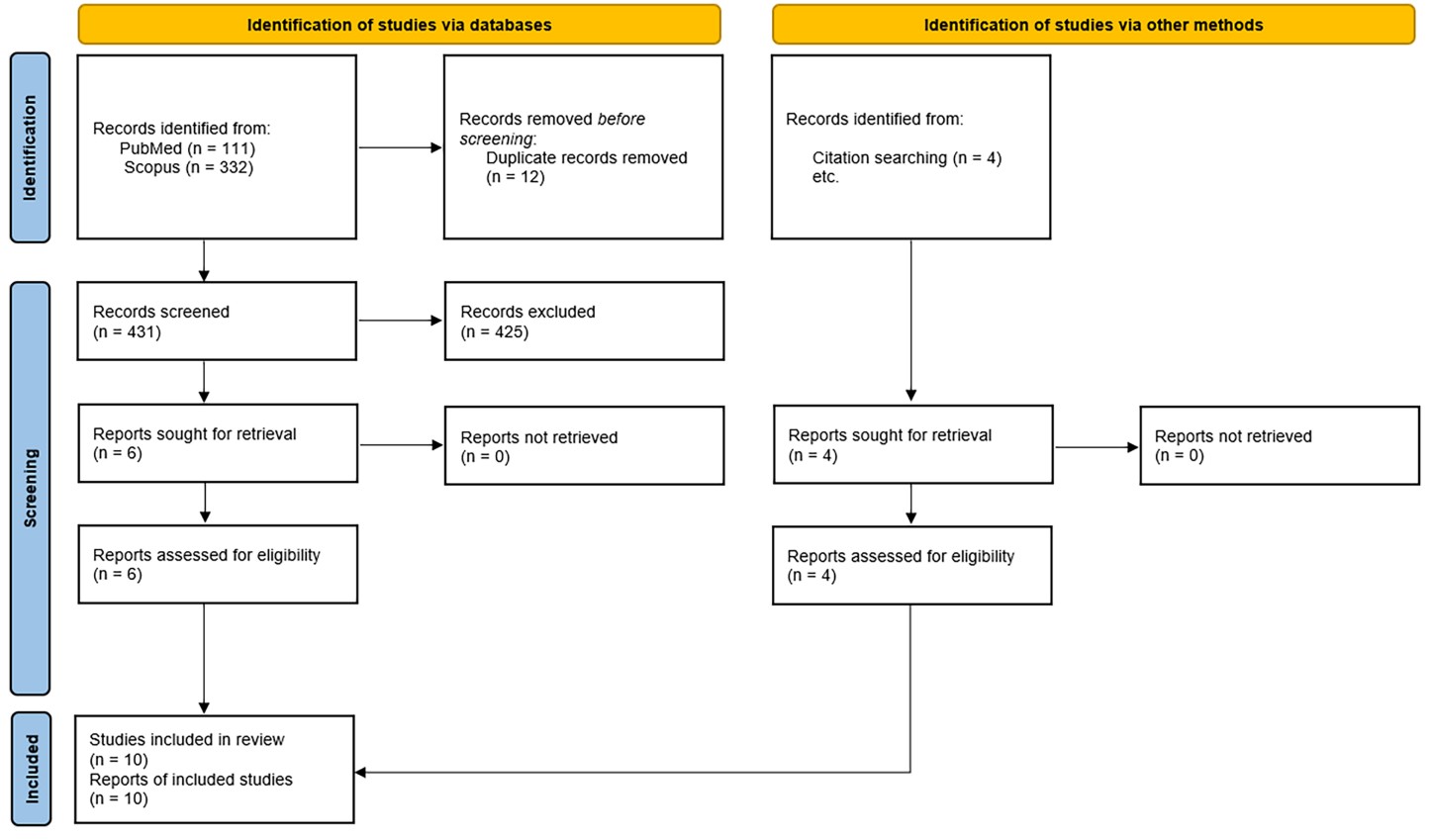

**Figure 1 Flow chart of studies identified for inclusion.**

**Table 3 Characteristics of included sporting tournaments.**

| Tournament | Outcome | Location of changed births | Tournament dates | Sport | Continent | Context |
|---|---|---|---|---|---|---|
| FIFA World Cup and UEFA Euros, European national teams (*Fumarco & Principe, 2021*) | Nine months following a performance improvement of one standard deviation, there was a 0.3% decline in births nine months on. | European country | 1960 to 2016 | Association football (soccer) | Europe | A unique dataset integrating metrics of national teams' performance in 27 international football competitions with monthly birth rates at the country level for 50 European countries during a 56-year period. |
| 1998 FIFA World Cup (*Masukume & Grech, 2016*) | The sex ratio at birth did not increase nine months after the tournament in France. | N/A | 10 June to 12 July 1998 | Association football (soccer) | Europe | This was the second time France (and ninth time Europe) hosted this tournament. This was France's first World Cup victory. |

| Tournament | Outcome | Location of changed births | Tournament dates | Sport | Continent | Context |
|---|---|---|---|---|---|---|
| Spanish major league (*la Liga*) (*Bernardi & Cozzani, 2021*) | Nine months after an unexpected loss by the most popular *la Liga* soccer team in a Spanish province, there were 0.8% fewer births there for the studied years 2001–2015. Unexpected wins did not alter the number of births. | Spain, provinces | 2000/2001 to 2014/2015 *la Liga* seasons | Association football (soccer) | Europe | *La Liga* is the top level of Spanish football leagues for men's professional football. |
| Super Bowl, NFL (*Grech & Zammit, 2019*) | ~Nine months following the first Sunday in February, there was an increased sex ratio at birth in 11/2006, 9/2009, 10/2009, 10/2010, 12/2010, 11/2011, 11/2013, and 12/2013.* | United States | 5 February 2006, 1 February 2009, 7 February 2010, 6 February 2011, 3 February 2013 | American football | North America | The Pittsburgh Steelers won in Michigan in 2006, Florida in 2009, the New Orleans Saints in 2010, the Green Bay Packers in 2011, and the Baltimore Ravens in Louisiana in 2013. |
| Super Bowl, NFL (*Hayward & Rybińska, 2017*) | There was no observable pattern of birth increases in winning counties nine months after the Super Bowl. In a similar vein, losing a Super Bowl was not associated with a changed birth pattern. | N/A | 1st Sunday of February 2004–2013 | American football | North America | In Texas in 2004 and Florida in 2005, the New England Patriots triumphed. In 2007, the Indianapolis Colts won in Florida. In Arizona in 2008 and Indiana in 2012, the New York Giants triumphed. |
| 2009 UEFA Champions League (*Montesinos et al., 2013*) | In February 2010, nine months following FC Barcelona's victory in May 2009, there was a 16% increase in births. | Counties of Solsonès and Bages in Catalonia, Spain | May 2009 | Association football (soccer) | Europe | FC Barcelona advanced to the finals thanks to a late goal by Andrés Iniesta. By winning this competition, which had its final in Italy, FC Barcelona became the first Spanish team to win the treble (national league, national cup, and continental trophy). |
| 2010 FIFA World Cup (*Masukume & Grech, 2015*; *Masukume, Grech & Scherb, 2016*) | In February and March 2011, nine months after the tournament, the sex ratio at birth increased, and there were over 1,000 additional births. | South Africa | 11 June to 11 July 2010 | Association football (soccer) | Africa | This was the first time South Africa (and Africa) hosted this tournament. South Africa scored the tournament's first goal, which was considered one of the best in the tournament. |
| 2016 UEFA Euros (*McKenna, Znaczko & Morrison, 2019*) | In March 2017, nine months after Northern Ireland's participation in the tournament, there was a 2% increase in births. | Northern Ireland | 10 June to 10 July 2016 | Association football (soccer) | Europe | This was the first time Northern Ireland qualified for the UEFA Euros. Northern Ireland advanced to the knockout stages of this tournament held in France. |

(Continued)

 

| Tournament | Outcome | Location of changed births | Tournament dates | Sport | Continent | Context |
|---|---|---|---|---|---|---|
| Table 3 (continued) | | | | | | |
| 2019 Rugby World Cup (*Inoue & Mizoue, 2022*) | In September 2020, 10 months after the tournament, there was an increase in the sex ratio at birth. | Japan, some prefectures | 20 September to 2 November 2019 | Rugby | Asia | This was the first time Japan (and Asia) hosted this tournament. Japan reached the quarterfinals for the first time, winning all four of their group matches. |

**Note:**
*Now played on 2nd Sunday of February since 2022; FIFA, *Fédération Internationale de Football Association*; UEFA, Union of European Football Associations; FC, Futbol club; NFL, National Football League; N/A, Not Applicable.

9(±1) months after notable team victories and/or the tournament's hosting. Related to this, a drop in birth rates nine months later was linked to a team from a top soccer league suffering unexpected losses (Table 3).

## DISCUSSION

### Principle findings

In this study, we found that five major sporting tournaments were linked to noticeably increased birth metrics 9(±1) months later. The Super Bowl (increased US birth sex ratio in multiple years (*Grech & Zammit, 2019*), however, from 2004 to 2013 there were no observable birth increases in winning counties and losing was not associated with a changed number of births (*Hayward & Rybińska, 2017*)), the 2009 UEFA Champions League (16% increase in Solsonès and Bages births in Spain (*Montesinos et al., 2013*)), the 2010 FIFA World Cup (increased birth sex ratio (*Masukume & Grech, 2015*) and over 1,000 extra births in South Africa (*Masukume, Grech & Scherb, 2016*)), the 2016 UEFA Euros (2% increase in Northern Ireland births (*McKenna, Znaczko & Morrison, 2019*)) and the 2019 Rugby World Cup (increased birth sex ratio in some Japanese prefectures (*Inoue & Mizoue, 2022*)). Nine months after the most popular provincial *la Liga* soccer teams unexpectedly lost matches, there were 0.8% fewer births in those provinces from 2001 to 2015; the number of births were unaffected by unexpected wins (*Bernardi & Cozzani, 2021*). After the 1998 FIFA World Cup a changed sex ratio at birth was not witnessed nine months on (*Masukume & Grech, 2016*). Nine months following a performance improvement of one standard deviation by a European national soccer team at the FIFA World Cup or UEFA Euro Championships, from 1960 to 2016, there was a 0.3% decline in births nine months on (*Fumarco & Principe, 2021*).

### Comparison with other studies

Some studies have linked a surge in births nine months later to several occasions, such as carnivals, religious events and secular holidays like Christmas and New Year's Day (January 1). This increase has been attributed to the holiday celebrations, which promoted greater conception through increased sexual activity among the public (*Jukic et al., 2013*; *Kadhel et al., 2017*; *Macfarlane et al., 2019*; *Régnier-Loilier & Rohrbasser, 2011*). Although

exceptions exist, the principle finding of this systematic review, of baby booms 9(±1) months after winning or hosting major sports tournaments, across different sports and continents, is consistent with the mechanism of celebratory sexual intercourse (*Hayward & Rybińska, 2017*). After a team wins, it has been hypothesized that a person's happiness coincides with a hormonal shift that elevates their desire for sexual activity (*Casto & Edwards, 2016*). Large-scale human emotions can therefore have a significant impact on population demographic changes.

Despite the fact that there was no general increase in live births after the Super Bowl (*Hayward & Rybińska, 2017*), the ratio of male to female live births increased nine months later (*Grech & Zammit, 2019*). Heightened alcohol and tobacco consumption during celebratory periods in the first trimester following Super Bowl victories has been associated with reduced birth weight nine months later (*Brian, Hani & Daniel, 2016*). Celebratory behaviour can manifest in various ways, encompassing both inebriation and heightened sexual activity (*McKenna, Znaczko & Morrison, 2019*). Thus reduced birth weights are not incompatible with increased male to female ratios at birth. While both the number of births and the male to female live birth ratio increased after the 2010 FIFA World Cup. This indicates that it is useful to consider both the overall number of live births and the births' male to female ratio (SRB).

Although England, Northern Ireland, Scotland and Wales are separate countries with their own national soccer teams, the United Kingdom was treated as a single entity in one investigation of this phenomenon (*Fumarco & Principe, 2021*). According to this investigation, there was a 0.3% drop in births nine months after a national team performance increase of one standard deviation. The 2016 UEFA tournament in which Northern Ireland participated, had a 2% increase in NI births nine months later, which appears to be in conflict with this investigation (*McKenna, Znaczko & Morrison, 2019*). The findings of this investigation (*Fumarco & Principe, 2021*) should thus be handled cautiously given how the United Kingdom was treated as a single nation.

The context in which baby booms take place appears to be important, such as when a nation hosts big international sporting events for the first time (*e.g.*, South Africa 2010 FIFA World Cup (*Masukume & Grech, 2015*; *Masukume, Grech & Scherb, 2016*), Japan 2019 Rugby World Cup (*Inoue & Mizoue, 2022*)), or when it first qualifies for one and does relatively well (Northern Ireland 2016 UEFA Euros (*McKenna, Znaczko & Morrison, 2019*)). It is conceivable that during these inaugural tournaments, public excitement is at its peak, increasing the frequency of celebratory sexual intercourse. This may help to explain why France, despite winning the tournament, did not experience a baby boom nine months after the 1998 FIFA World Cup as it was the country's second time hosting the tournament and it had previously qualified (*Masukume & Grech, 2016*). Unexpected defeats in the Spanish top division *la Liga*, when fewer births were recorded nine months later, are another contextual element (*Bernardi & Cozzani, 2021*). An unexpected defeat is more likely to have a negative impact on mood than an expected defeat, with probable aftereffects including a reduction in the number of sexual encounters and subsequent births nine months later.

That a change in the number of babies born or in the sex ratio at birth occurred *circa* nine months after major sporting tournaments across different sports and continents suggests a true effect and a common underlying mechanism. This mechanism suggests that euphoria or sadness experienced when a sports team wins or loses coincides with a hormonal change that increases or decreases a person's desire for sexual activity, which then affects birth rates nine months later. It appears that major multi-sport events like the Olympic Games do not lead to changed birth rates nine months later (*McCartney et al., 2010*). In contrast, this systematic review suggests that major single-sport events, such as those in soccer, rugby, and football league, featuring a singular apex game or tournament, likely generate more focused attention and heightened excitement, contributing to observable effects on birth rates. Baby booms post major single-sport events can potentially shape social norms, influencing family planning and creating unique cultural narratives, prompting future exploration.

### Strengths and limitations

A noteworthy strength is the methodology, which is transparent and replicable, and does not impose any language restrictions on the search. Our investigation was constrained because we could have overlooked important articles by not searching gray literature. Also, potential selective reporting of significant findings remains a concern. This study has the same drawbacks as ecological studies, such as its restricted capacity to draw conclusions at the individual level (the ecological fallacy) (*Björk et al., 2021*). However, it is probable that ecological research is one of the best ways to assess how a population event affects a population outcome (*Pearce, 2011*).

## CONCLUSIONS

With a few exceptions, major American football, Association football (soccer) and Rugby apex tournaments in Africa, North America, Asia and Europe were associated with increases in the number of babies born and/or in the birth sex ratio 9(±1) months following notable team wins and/or hosting the tournament. Related to this, unexpected losses by teams from a premier soccer league were associated with a decline in births nine months on. The systematic review's implications span healthcare planning and policy, prompting consideration of major sporting events as birth pattern and sex ratio influencers, which may subsequently influence the demand for midwives, doctors, and healthcare resources. This underscores societal events' potential in shaping demographic trends and invites future research to uncover underlying mechanisms and cross-cultural applicability. In light of the evolving scale and scope of major sporting events, such as the upcoming 2026 FIFA World Cup with increased teams and hosts, further research is warranted to provide nuanced estimates and boundaries for the impact on child birth rates, acknowledging the changing dynamics of these tournaments (*Beissel & Kohe, 2022*). In conclusion, Nelson Mandela was correct when he averred, "Sport has the power to change the world. It has the power to inspire, it has the power to unite people in a way that little else does." (*Nelson Mandela Foundation, 2000*).

### Funding

The authors received no funding for this work.

### Competing Interests

The authors declare that they have no competing interests.

### Author Contributions

- Gwinyai Masukume conceived and designed the experiments, performed the experiments, analyzed the data, prepared figures and/or tables, authored or reviewed drafts of the article, and approved the final draft.
- Victor Grech conceived and designed the experiments, analyzed the data, authored or reviewed drafts of the article, and approved the final draft.
- Margaret Ryan conceived and designed the experiments, performed the experiments, analyzed the data, authored or reviewed drafts of the article, and approved the final draft.

### Human Ethics

The following information was supplied relating to ethical approvals (*i.e.*, approving body and any reference numbers):

n/a.

### Data Availability

The Scopus and PubMed search results are available in the Supplemental File.

### Supplemental Information

Supplemental information for this article can be found online at http://dx.doi.org/10.7717/peerj.16993#supplemental-information.

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
