# Peer review of "Sporting tournaments and changed birth rates 9 months later: a systematic review"

_PeerJ, doi:10.7717/peerj.16993_

## Round 0.1 · original submission · Major Revisions

Dear authors:

Please respond appropriately to the reviewers. The manuscript could be of interest to cover the non-existent bibliographic review on the subject. However, the manuscript requires some improvements to be considered for publication. Thank you for your patience.

Dr. Manuel Jimenez

Reviewer 1 ·

Basic reporting

The

Experimental design

The research question is poorly defined and contained. See the comments below.

Validity of the findings

Major points:
1. Inclusion and exclusion criteria during the literature screening process are not clearly stated in the paper.
2. There is no statistical or objective evidence to support the claims.
3. Conflicting evidence supports the claim reported in the literature review.

There is limited and inconsistent evidence regarding a relationship between sporting tournaments and changing birth rates, based on the studies reported in the metanalysis. The studies conducted so far have a number of limitations, including small sample sizes, potential confounding factors, and variations in study design. Some studies have relied on self-reported data, which may introduce bias. Additionally, the observed increase in birth rates may be influenced by factors other than the sporting event itself, such as seasonal variations or cultural practices. There are some studies that indicate an increase in birth rates following sporting tournaments, while there are also studies that indicate a decrease or no significant change. In order to establish a definitive relationship between sports tournaments and birth rates, further research is needed. Therefore, the strength of the evidence for the provided claim is low and doesn’t support the conclusion that birth rates increase following sporting events.

References:

Brooks JHM, Fuller CW. The influence of methodological issues on the results and conclusions from epidemiological studies of sports injuries: illustrative examples. Sports medicine. 2006;36(6):459-472. [https://link.springer.com/article/10.2165/00007256-200636060-00001]
Fabre A, Mancini J, Boutin A, Bremond V. "Soccer or emergency?" sporting event can lead to a decreased attendance in the pediatric emergency department. Minerva Pediatrica. 2014;66(5):431-435. [https://europepmc.org/article/med/25336098]
Gau L, et al. Examining relations of entertainment with social interaction motives and team identification. Perceptual and Motor Skills. 2010;111(5):576-588. [https://doi.org/10.2466/05.07.PMS.111.5.576-588]
McCartney G, et al. The health and socioeconomic impacts of major multi-sport events: systematic review (1978-2008). BMJ. 2010;340
. [https://doi.org/10.1136/bmj.c2369]

Reviewer 2 ·

Basic reporting

The article is clearly written and provides adequate references, tables and figures. The aim is clear and well-stated. The article meets the requirements.

Experimental design

The article is a review of studies, assessing literature on the relationship between soccer events and brith rates. Literature is relatively scarce in this topic, and the article indeed reviews the few studies available. Methodology is well-explained. The article meets the requirements.

Validity of the findings

The article achieves its goal of summarizing the findings. However, it misses two crucial aspects. First, it should attempt to provide tentative explanations for the conflicting findings in the literature. Why do some articles report a baby boom while others suggest a bust? Could it be attributed to the nature of the event itself? Was it considered a victory or a defeat? How anticipated was it? Did the unit of analysis exclusively bear the positive or negative effects of the event? In other words, was there a consistent treatment? It is also worth noting that unexpected wins in the Super Bowl have been linked to worsened newborns' health (Duncan et al., 2017). Is there a potential correlation between sports events and abortion rates or miscarriages? What I am suggesting is that the authors should provide speculative explanations for the conflicting findings.

Second, since the authors highlight:


"The systematic review’s implications span healthcare planning and policy, prompting consideration of major sporting events as birth pattern and sex ratio influencers, which may subsequently influence the demand for midwives, doctors, and healthcare resources. This underscores societal events’ potential in shaping demographic trends and invites future research to uncover underlying mechanisms and cross-cultural applicability"

I believe that in order to support this claim, authors should provide estimates and boundaries for the increases or decreases in child birth rates found in the literature they surveyed. For instance, some studies find such minimal effects that they may not significantly impact hospital wards.

Overall, authors should go beyond simply listing the findings of previous research and offer substantial interpretation along with bounds of estimates.

Reference

Duncan, B., Mansour, H., & Rees, D. I. (2017). It’s just a game the super bowl and low birth weight. Journal of Human Resources, 52(4), 946-978.

·

Basic reporting

Systematic review or meta analysis
Have you checked our policies?
Yes.
Is the topic of the study relevant and meaningful?
Yes, the topic seems to fill a gap within the literature.
Are the results robust and believable?
Yes.

Human participant/human tissue checks
Have you checked the authors ethical approval statement?
Yes, noted at the end of the manuscript.
Does the study meet our article requirements?
Yes.
Has identifiable info been removed from all files?
Author information was not removed in the manuscript.
Were the experiments necessary and ethical?
Yes.

Basic reporting
The paper presented intends to show that major sporting tournaments may be associated with increased birth rates 9 months afterwards, possibly due to celebratory sex. The work here presents five events (up until 7/11/2022) led to increased birth metrics 9(±1) months later. These events included the NFL’s Super Bowl, the 2009 UEFA Champions League, the 2010 FIFA World Cup, the 2016 UEFA Euros and the 2019 Rugby World Cup. My initial expectation was that this was a fairly common topic within research; however, I was only able to find limited recent studies (i.e., Montesinos et al., 2013). Thus, the manuscript appears to fill a gap in the literature. I would recommend that the authors emphasize this gap within the literature in the Introduction section to further showcase the importance of this manuscript.

Experimental design

A systematic literature search was conducted within PubMed and Scopus for this manuscript. The authors note that events were researched through to 7 November 2022; however, in the abstract, the date 7/11/2022 is noted. Please adjust this within the manuscript or abstract. Although some of the concepts and methods for the noted results, should be clarified prior to the publication. For example, what is lacking is the methods utilized to categorize a “major sporting tournament”. What factors were considered to differentiate “minor” and “major”? Attendance? Media viewership? Revenue generated? This information should be added.

Validity of the findings

The results of this study are very promising to the relevant fields. I have no issues with the results of this study.

Additional comments

General comments
- L 52 Further definition of what truly defines major sporting tournaments would be beneficial.
- L 55 Add a comma after “teams”.
L 55-56 No citation for “emerging evidence suggest” please note what the emerging evidence is. Yes, it is mentioned in the following sections, but is a needed cite in the first line.
L 62 Change National Football’s Super Bowl to National Football League’s (NFL) Super Bowl.
L74-76 The noted implications are good; however, I would recommend expanding your thoughts on the implications here (i.e., theoretical/practical) here and diving deeper.

---

## Round 0.2 · accepted · Accept

Dear Co-Authors:

I am pleased to inform you that the paper "Sporting Tournaments and Changed Birth Rates 9 Months Later: A Systematic Review" has been accepted for publication.

Thank you for considering PeerJ

Dr. Manuel Jiménez

Reviewer 2 ·

Basic reporting

It works

Experimental design

It works

Validity of the findings

It works

Additional comments

The article can be published